# "The Impersonal You": Mass Print and Other Communication Technologies in the Virtual Friendship of Harriet Beecher Stowe and George Eliot

**Olga Kuminova** 

Abrahams-Curiel Department of Foreign Literatures and Linguistics, Ben-Gurion University of the Negev, P.O. Box 653, Beer Sheva 84105, Israel; kolga@post.bgu.ac.il or olga.kuminova@gmail.com; Tel.: +972-52-4439877

**Abstract:** The relationship between Harriet Beecher Stowe and George Eliot, widely recognized as one of the most significant literary friendships in the 19th century, yet rarely focused on in scholarship beyond mutual literary influence, took place entirely through the communicative media available then: mass print, the Victorian post, and the social network of parlor literature and transatlantic literary community. The article analyzes the beginning of the correspondence, both similar to and different from fan mail exchange, with extensive quotes from Stowe's unpublished second letter, to demonstrate an innovative theoretical point that novels can function as part of a communicative continuum between a writer and an individual reader, becoming instruments of what may be seen as a proto-virtual relationship.

**Keywords:** Harriet Beecher Stowe; George Eliot; correspondence; parlor literature; mass print; fan mail; novel; written communicative gesture; transatlantic; virtual friendship

## 1. Introduction

In the 1980s and 1990s, a new perspective was created on the British and American literary history of the long 19th century: the newly authoritative critical approaches of feminist criticism, history of reading and new historicism converged in the seminal works of such scholars as Jane Tompkins, Nancy Armstrong, and Jennifer Phegley to reconstruct the crucial cultural work performed by texts circulated between women writers and women readers of that period on both sides of the Atlantic.[1] Writers such as Harriet Beecher Stowe, Susan Warner, or Elizabeth Stuart Phelps, tremendously popular and influential in their own lifetime, were redeemed by this new scholarship from the status of sentimental (invariably connoting "inferior") fiction to which they had been relegated by the modernist, pointedly apolitical and predominantly masculine critical-academic establishment.

In particular, Armstrong (1987) argues in *Desire and Domestic Fiction* that the emotional, intellectual, and erotic subjectivity of female characters, constructed by domestic fiction as independent of class and social status, made a major contribution to the emergence of middle-class power, concurrent with the rise of the novel. In other words, opposing the modernist dismissal of domestic fiction, Armstrong presented women characters and women writers as key agents in forging a new social contract that is still with us today. The present article presents a similarly conceived argument against another modernist attitude, deeply related to the one Armstrong opposes—a distrust of mass print

---

[1]  (Tompkins 1985) *Sensational Designs: The Cultural Work of American Fiction, 1790–1860*, (Armstrong 1987) *Desire and Domestic Fiction: A Political History of the Novel*, (Phegley 2004) *Educating the Proper Woman Reader: Victorian Family Literary Magazines and the Cultural Health of the Nation*.

and broader, ever-developing communicative media as intrinsically inferior and "cheap", eroding the unique individual authenticity and value of both the work and the reader/writer.

Everyone is familiar with the spirit of Walter Benjamin's lament over the nearly-extinct, privileged authenticity of encountering a unique work of art in the original. In "The Work of Art in the Age of Mechanical Reproduction", Benjamin points out "the contemporary decay of the aura. It rests on two circumstances, both of which are related to the increasing significance of the masses in contemporary life. Namely, the desire of contemporary masses to bring things "closer" spatially and humanly, which is just as ardent as their bent toward overcoming the uniqueness of every reality by accepting its reproduction" (Benjamin 1969). This opposition between technological progress, epitomized by mass (re)production of art objects or literary texts, and authentic artistic experience, representing unique subjective meaning in general, emerged and increased as the sizes of art and literature audiences and the numbers of artists and writers themselves, as well as the works they produced, kept growing by orders of magnitude since the beginning of the industrial revolution and the concomitant "printing" and "reading revolution".

Sixty years before Benjamin's essay, George Eliot, an eminently successful publishing author in the age of growing expansion of mass print, expressed a related, though not identical concern. In a journal entry from 13 January 1875, she wrote about her "fear lest I may not be able to complete it [*Daniel Deronda*] so as to make a contribution to literature and not a mere addition to the heap of books"[2]. Eliot's anxiety about the unique significance of her novel in progress may seem paradoxical in such a mature and established writer. Yet it is in line with contemporary perceptions of the industry of mass print as one of the alienating capitalist institutions that blight the individual's perception of her life and her work as uniquely, authentically meaningful. I suggest that Eliot's concern has to do not only with the *quality* of her work, but also with the sheer *quantity* of the printed matter churned out by the press, in which both the reader and the individual work would be likely to drown, unable to assign or produce unique individual meaning. A paradigmatic contemporary example of rebellion against this perceived tendency in book production was William Morris's Kelmscott Press, where he strove to make each book, literally each copy of it, sufficiently unique to matter (Henderson 1973).

This article aims at demonstrating how, in spite of this seeming contradiction or conflict, the developing media of communication, such as mass print, fast and efficient post and even transatlantic telegraph (as an existing if not actualized possibility) *enabled*, rather than hindered or erased, the production of unique subjective meaning for books, their writers, and their individual readers.

## 2. The Correspondence: Publications and Scholarship

Although Harriet Beecher Stowe visited Britain in 1853 with a triumphant lecture tour following the instant popularity of *Uncle Tom's Cabin*, she was not at that time aware of George Eliot's work, whose first landmark novel *Adam Bede* came out only in 1859. After Stowe first wrote to Eliot in April 1869 (Beecher Stowe 1869a), she repeatedly invited her British correspondent to visit her in the United States. Yet, in spite of expressing regret about it, Eliot never did, whether because of her own and her family's health problems, or because of her "scandalous" marital status that would not allow her to be unproblematically accepted by the public, or merely because she had other priorities. Nevertheless, Eliot maintained the connection, which took the form of correspondence, mutual commentary on each other's work, and mutual literary influence[3] until her death in 1880.

---

2    Cited by Eliot's biographer Rosemary Bodenheimer in p. 175. Generally, Bodenheimer points out that "[b]eginning in the late 1860s and continuing through the writing of *Middlemarch* and *Daniel Deronda*, the letters express a new phase of anxiety about her own writing and a renewed intensity of criticism concerning the production and circulation of bad literature. Her suffering now took the form of testing herself against her earlier work and fears of repetition or "excessive writing"" (p. 174).

3    For a discussion of such influence, see, e.g., Hack (2013) "Transatlantic Eliot: African American Connections," tracing the influence of Stowe's *Dred* on *Daniel Deronda.*

This correspondence has never been published in full. All of Eliot's letters to Stowe are available in Gordon Haight's seven-volume collection of her letters, where only one letter's beginning is lost. Three of Stowe's letters to Eliot were published, with substantial editing, in her (Stowe (1889)) biography by her son Charles Edward Stowe, and five reproduced with abridgments in Fields' (1897) biography of Stowe. Four appeared in full with brief analytic introductory commentary by Jennifer Cognard-Black in a collection of letters by women writers entitled *Kindred Hands* (Cognard-Black and Walls 2006). The fragments of Stowe's letters that comment on Eliot's work are fairly well known and often quoted by literary critics. Six of Stowe's letters remain unpublished and are stored in the Berg Collection of New York Public Library[4].

Up until recently, there has been relatively little scholarship devoted to this remarkable epistolary friendship. Researchers often refer to the correspondence in passing, quoting it for specific biographical information or samples of critical commentary, especially that by Stowe on *Daniel Deronda*, which is an important critical piece, unpublished at the time (nor was it reproduced by Charles E. Stowe or Fields)[5]. The talk of Margaret Wolfit at the George Eliot Birthday Luncheon, 20 November 1988, was composed of chronologically arranged fragments, including much of the above critical commentary and designed to briefly introduce the correspondence.[6] Yet scholarly attention rarely focuses on this correspondence as a primary text, as, according to Rosemarie Bodenheimer, one must do if one wants to engage in letter criticism, which is not supposed to "privilege the letter fragment" as truth, fact, or evidence, but interpret it in context as a textual construct in its own right (Bodenheimer 1994, p. 7).

There is a small number of publications that focus more closely on this correspondence as a whole: first, Marlene Springer's detailed and informative essay (1986) restored, for the most part, the order of the correspondence and summarized its content with some extensive quotes, while providing biographical background. Almost two decades later, Jennifer Cognard-Black's introduction to the above-mentioned publication of four of Stowe's letters in *Kindred Hands* emphasized the feminine solidarity and mutual support of the two writers and their shared belief "in the power of artistic femininity to shape national culture" (Cognard-Black and Walls (2006), p. 23). Cognard-Black also made the correspondence one of her central sources in *Narrative in the Professional Age,* where the interaction between Stowe and Eliot is extensively and thoroughly analyzed in terms of shaping their professional identity as women writers in the national and transatlantic contexts. In her 2013 book *Poetics of Character: Transatlantic Encounters 1700-1900,* Susan Manning offered a subtle and powerful, though brief analysis of the two writers' relationship from the point of view of American literature's political "anxiety of influence". Most recently, Midorikawa and Sweeney (2017) undertook a massive effort of familiarizing the public with this literary friendship, in their narrative account of the correspondence in the 2017 book *A Secret Sisterhood: The Literary Friendships of Jane Austen, Charlotte Brontë, George Eliot, and Virginia Woolf,* and a number of items that followed in the press and on YouTube.

## 3. Women, Epistolary Culture, and the Post

Some background on female letter writing in the nineteenth-century provides a useful context for analyzing the correspondence. Letter writing by women was, on the one hand, encouraged as a useful exercise that would prevent the mind from staying idle and would foster connections with family and friends. On the other hand, many commentators and authors of letter-writing manuals also feared female correspondence to some extent (as they did novel reading): if women, especially young unmarried ones, indulged too much in the pleasures of private exchange of letters, it might lead to all kinds of problems, from exposing family secrets to sinful idleness. Even the champion of women's

---

[4] One, presumably short letter known to be missing was inscribed by Stowe in a copy of *Uncle Tom's Cabin* she sent as a gift to Eliot, which was sold at Sotheby's in 1923, lot # 561 (Haight 1954–78, Volume VII, p. 132; Haight notes that the short letter referred to the book and the abolition of slavery).

[5] e.g., Joan Hedrick, *Harriet Beecher Stowe: A Life* (Hedrick 1994).

[6] (Wolfit 1989) "The Toast to the Immortal Memory", *George Eliot Fellowship Review.*

education Hannah More denounced the "excessive mutual flattery" that, in her opinion, characterized young women as correspondents (Bodenheimer 1994, chp. 1). This might be interpreted as moralistic concern over young women consuming too rich emotional nourishment.

On the whole, the epistolary medium itself was increasingly feminized in Western culture stereotypically—as opposed to men who write such public texts as books, newspaper articles, and business, legal or state documents, women write private letters[7]. However, both Jane Gallop and Steedman (1999) examine letters by women as a site of the emergence and manifestation of women's subjectivity. Rosenberg (1975) provides a valuable social context for making sense of the degree of affection manifested in nineteenth-century letters between American women: written according to the conventions of the relatively segregated, homosocial female "world," these letters generally expressed affection in highly emphatic ways. Rosenberg's analysis suggests that Stowe's highly affectionate language in the letters to Eliot should not be seen as exceptional (and Eliot's more reserved tone may say more about cultural differences than anything about a lack of reciprocity in their relationship).

In both England and America, letter writing was closely associated with duties and obligations, whether professional or personal, and was a demanding part of one's daily routine. Within London, for instance, business etiquette required a regular letter to be answered within one or two post collections after receipt, according to Golden (2009). This would give the receiver between 1-4 hours to reply—strikingly, about the same time within which an answer to an email is expected today. Yet in the correspondence between Stowe and Eliot, there were intervals where one or two years would pass between a letter and a reply—and still, the reply remained possible—which testifies to the strength of the connection between them, the mechanisms of which I will further attempt to understand.

Both correspondents were aware of the drain that daily correspondence puts upon the other's writing time and energies, and each one, more than once, expressed concern not to overtax the other with this correspondence. This stands in an apparently paradoxical opposition to their explicitly expressed view of their correspondence as an important personal resource. They also shared a sense that published work provided a complementary, perhaps even the central, channel of communication between them: as Stowe writes in 1869, "you have said so much in books that you *need* not to write me letters." Yet the fabric of obligations, time limits, caution, etiquette, and apology informs even the most informal, warm, and open of their letters.

## 4. Novels and Letters: One Communicative Continuum

In my earlier article, I discussed the situation of a writer, more specifically a novelist, in the age of mass print and the reading revolution in eighteenth-century Europe and Britain (Kuminova 2011). I argued that writing for an undefined, anonymous, and broadening audience presented a specific challenge to writers, which they coped with in various ways, particularly by embedding the narrative within fictional relationships and communicative contexts (as in epistolary novels, fictional diaries, or memoirs), crafting paratexts such as prefaces directly addressing the readers, or even addressing the "dear reader" in the body of the narrative. In addition to this challenge posed by the technical nature of the medium, the economics of mass market raised concerns about determining—or producing—literary value.

George Eliot's anxiety that her novel might be swallowed up by the monstrous heap produced by the press appears to have a similar genesis. To combat the impersonality of mass print addressed to an unknown mass audience, Eliot had her own ways of re-personalizing literary communication—particularly, her insistence on the distinctive quality and originality (unique significance) of her writing, and her view of reception as stratified. She believed that the popularity and influence of significant books spread gradually by way of interpersonal communication, starting from the more competent readers: "if a book which has any sort of exquisiteness happens also to be a

---

[7] For cultural critique of such feminization see e.g., (Gallop 1985).

popular, widely circulated book, the power over the social mind for any good is, after all, due to its reception by a few appreciative natures, and is the slow result of radiation from that narrow circle" (Haight 1954–78, pp. 30–31)[8]. In other words, it is not the printing press that propagates a worthwhile book, and not the market that establishes its value—it happens by no means mechanically.

As for Stowe, who started writing in the framework of parlor literature, her literary works began in the modality of personal letters to the reader, addressed within a relationship of social closeness and moral common ground. Parlor literature, a concept introduced by Stowe's biographer Hedrick (1992), encompassed the "domestic" literary forms written and read by women in literate American households: letters, verse, essays for literary clubs, all of which were impossible to separate from the fabric of conversation in the parlor. Hedrick quotes Elizabeth Cady Stanton, who listed "the parlor, press, and pulpit" as three equally important arenas of public opinion. (Catherine Allgor extends this notion to "parlor politics," the informal avenues of women's influence in American politics of the time)[9]. Phegley (2004) extensively discusses the role of "family magazines," which brought a range of American and British literature and criticism in an attractive format and at affordable prices to precisely this audience of women in the parlor.

That Stowe did not perceive the publishing industry and journal criticism of literature as operating impersonally, even after achieving truly mass print runs, is evident from such comments as the one in her letter to E.L. Godkin, the editor of *Nation,* following a vitriolic and misogynist review of *Oldtown Folks* in his journal. Stowe admonished Godkin (who was her relative through marriage and occasionally visited the Stowes at home; (Hedrick 1994, p. 347): "Rudeness and heedless discourtesy, an assumption of supercilious contempt I think are just as much out of place in literary criticism as in a private parlor"[10]. To the note of apology that he sent in reply, she responded: "Accidents will happen you know in the best regulated fam[ilie]s"[11], drawing an implicit parallel between a publishing enterprise and a "well-regulated," genteel family. Moreover, Stowe's relationship with her main publisher, James Thomas Fields, was not compartmentalized as a strictly professional one. Due mostly to the social talents and initiative of Annie Adams Fields, who kept a literary salon, the Fields' house in Boston was a site of informal, not exclusively professional interaction for many American writers—which kept the "publishing industry", in any case as represented by James Thomas and Annie Adams Fields, from being completely and solely a mechanical monster, as it was often perceived at the time.[12] Annie Fields herself was one of Stowe's closest friends and a mutual acquaintance who created a bridge between her and George Eliot.

In the course of their correspondence, Stowe continued to use Eliot's newly published literary work as additional, though more elaborate, more thematically focused, and less personal, "letters" from the author, to which she responded in her own letters. In 1876, when *Daniel Deronda* started being published in monthly installments in *Harper's Weekly*, Stowe writes: "And now I am beginning to hear from you every month in Harper [sic]—& it is as good as a letter." (HBS11, *The Life*). A syntactic confusion in Stowe's second letter is revealing from this perspective. Stowe writes "You will see when you read [the word "this" crossed out] my book that much as you put heart & soul into Romola I have put heart and soul & lifeblood into this book [*Oldtown Folks*] which cost me more to write than anything I ever wrote" (Beecher Stowe 1869b). The confusion between "this", i.e., this letter, and her book, a confusion which she subsequently corrects by crossing out the word "this", is telling—it shows

---

8   For a recent discussion of Eliot's view of reception as stratified, see Alicia Williams "The Politics of Address in George Eliot's Fiction" (Williams (2019)).

9   For broader discussion of the political impact of American women's ostensibly private social and epistolary interactions, see (Allgor (2000)). On Stowe's narrators' personal address to the reader, see also Robyn Warhol "Letters and Novels 'One Woman Wrote to Another': George Eliot's Responses to Elizabeth Gaskell" (Warhol (1986a)), and "Toward a Theory of Engaging Narrator: Earnest Interventions in Gaskell, Stowe, and Eliot" (Warhol (1986b)).

10   HBS to E.L. Godkin, draft [Summer 1869], folder 256, Beecher Stowe Collection, Schlesinger Library, quoted in Hedrick (1994).

11   HBS to Annie Fields, 9 June 1869 (misdated by Stowe May 9), Fields papers, HL, quoted by Hedrick, ibid.

12   "The indefatigable press with its bottomless 'maw' [in Frank Norris' words] was figured as a kind of sorcerer's apprentice, with a near-demonic life of its own" (Hochman 2001, p. 25).

that for Stowe these two bodies of writing (letter and book) are in some sense initially equivalent and interchangeable.

A letter by Stowe to Eliza Follen (16 December 1852) is a similar kind of a message, in this case, privately addressed but written in the framework of preparations for Stowe's public visit to England[13]. As Susan Belasco notes, the letter soon ended up being "copied and reprinted in newspapers and magazines, in advance of her ... trip to England in 1853. The letter became famous as a major source of information about Stowe's life and was widely used in books and articles about her" (ibid.). Thus, many forms of writing—a private letter in response to a book (which might first be heavily annotated in pencil), manuscripts sent to a publisher who is also a friend, or private letters printed in a periodical, can be viewed as a communicative continuum that existed between writers, readers, and publishers, the interpersonal context of literary production.

## 5. The Stowe–Eliot Correspondence in the Context of Earlier and Contemporary Fan Mail

While, on the one hand, this correspondence is a unique record of a relationship between two exceptional, hardly typical women, on the other, it opens a perspective upon nineteenth-century fan mail, conventionally termed so, even though the term did not emerge until the early 20th century. The process of reading and its connection to subsequent correspondence with the writer that may ensue (writing a fan letter, receiving a reply, sometimes continuing the correspondence) can be viewed as a proto-virtual relationship developing in a combination of textual media (print and personal letter) enabled by new technologies of communication (mass print, including the publishing and distribution industries, and the efficient and cheap postal system). The phenomenon of a reader responding in a personal letter to the author of a literary text that touched him/her in an exceptional way arose more or less simultaneously with the mass-printed novel in the 18th century, when readers began writing to Samuel Richardson and Jean-Jacques Rousseau, often without prior personal acquaintance with the author[14]. Most researchers of nineteenth- and early twentieth-century fan mail address, in one way or another, the role that such text-mediated relationships with authors played in the reader's life and sense of self[15].

I would argue that the model of fan letter and response provides some insight into Stowe and Eliot's epistolary relationship. Although a famous writer herself, it appears that Stowe plays primarily the role of an enthusiastic reader and fan mail writer in her exchange with Eliot, while her correspondent mostly responds to topics raised and feelings expressed by Stowe. At the same time, of course, they are useful to each other as publishing writers, intellectuals, and supportive friends in this correspondence, as Springer and Cognard-Black note. Certainly, the power relations between these two writers are different from the ones that are supposed to characterize the communicative situation of a fan letter sent by an obscure, anonymous, "common" or non-professional reader to the author whose superior gift has been proved by public acclaim within nineteenth-century literary celebrity culture[16]. In addition, Manning (2013) places this correspondence against the complex background of political and literary anxiety of influence that afflicted American literature with regard to its British counterpart.

It must be immediately qualified, however, that what can be crudely phrased as the "common" reader looking up to the "great" figure of the writer is only one way of describing the communicative

---

[13]  See Belasco (2010), p. 62.

[14]  Robert Darnton mentions sackfuls of fan mail to Jean-Jacques Rousseau in "Readers Respond to Rousseau: The Fabrication of Romantic Sensitivity" (Darnton 1984, pp. 215–56). On readers writing to Samuel Richardson, see, e.g., Elspeth Knights, "Daring but to Touch the Hem of her Garment: Women Reading *Clarissa*" (Knights (2000)).

[15]  The publications on fan mail in this period include Susan Williams (cited above) and Brady (2011) on fan mail to Susan Warner; Robertson (2008) on the fans of Walt Whitman; Eisner (2007) on Elisabeth Barrett Browning as both a fan herself and fan mail receiver. Sources on early twentieth-century fan mail include Satterwhite (2011) on fan mail to local-color writers about the Appalachia; Barbara Ryan on readers' letters to Gene Stratton-Porter (Ryan (2002)); Jennifer Parcheski on Dorothy Canfield's readers (Parcheski (2002)); Courtney A. Bates on fan letters to Willa Cather (Bates (2011)).

[16]  See Rempe (2009).

situation of the fan letter, and it is certainly reductive, insofar as it presumes as given the very situation that the fan letter already sets out to transcend: writing a fan letter realizes the fan's strong and intrinsically egalitarian sense of relatedness to the author, the fan's stake in the author's mental universe and her right to her own written voice[17]. Moreover, as Courtney A. Bates notes in her discussion of fan letters to Willa Cather (and as examples provided by many other fan mail researchers demonstrate), fan letters were often written by readers who engaged with literature professionally—teachers, academics, aspiring or even established writers—thus narrowing the presumed divide between professional and unprofessional readers/writers.

## 6. The Initial Exchange of Letters: The Rhetorical Constructions of Virtual Friendship

Harriet Beecher Stowe's first letter to George Eliot is dated 15 April 1869 and sent from Mandarin, Florida, the Stowes' new winter home surrounded by an orange grove they planted, a setting that Stowe found very inspiring and energizing, as reflected, in particular, in the travel literature she wrote about Florida[18]. Stowe addresses Eliot as "My Dear Friend," although they never met or exchanged letters before. As one would expect after such an unconventional salutation, given that they are unacquainted, Stowe starts by explaining to Eliot how and why she decided to approach her directly. Of course, they had heard about each other, and Stowe even mentions in the letter's opening that Annie Fields had conveyed a greeting from Eliot to her a year ago; but she admits that bringing herself to initiate direct epistolary contact has taken her a whole year since she first felt a desire to do so: "Instead of writing to you, at that time I took Silas Marner and re-read carefully pencil in hand & then the Mill on the Floss. Then ... Adam Bede and then Romola—I have studied all these more than read them—& you will therefore see why it is that I must begin a note to you 'My dear friend'—" (HBS1, Cognard-Black 25). Stowe clarifies from the outset that studying Eliot's novels has been equivalent to her becoming intimately acquainted with the writer herself. Thus, the first letter can be seen as both continuing and "consummating" a previous one-sided communication that Stowe conducted with Eliot through the reading of her novels.

When people correspond by letter, the correspondents' pragmatic gestures are directed first of all at controlling or guiding the reception of the letter, overcoming the communicative obstacle of time and distance incorporated in its writing and sending (see, in particular, Elizabeth Hewitt)[19]. Stowe starts her first letter by outlining the pragmatic circumstances of writing it. She says it took her a long time to begin because she had been occupied finishing her latest book *Oldtown Folks*—"I was at that time heavily taxed writing a story that I am just now with fear & trembling giving to the English world[.] It is so intensely American that I fear it may not out of my country be understood, but I cast it like a waif on the waters" (HBS1). After outlining her predicament as a writer addressing an unfamiliar audience across the ocean, not unlike the precarious communicative situation of her present letter, Stowe assures Eliot that her addressee's previous messages (novels) did find a hospitable, sympathetic reception.

The next communicative gesture Stowe makes immediately after declaring and explaining her readerly friendship with Eliot is to give some carefully phrased writer-to-writer advice, although it might, in a fan letter proper, be merely a fan reader's wish: she thinks that Eliot should develop more extensively some of the stories sketched in *Scenes from Clerical Life.* (This gesture is paralleled later in the same letter with cautious criticism of Eliot's recently published long poem, "The Spanish Gypsy"—Stowe suggests that Eliot writes much better, "exquisite poetry in [her] prose"—without directly attributing any deficiency to Eliot as a poet, but making instead a polite generalization that

---

[17]  See, e.g., the discussion of the letters of Ann Gilchrist and other strong fans of Walt Whitman in Robertson's book.
[18]  See especially Eacker (1998). For the complex political connotations of owning an orange plantation in the post-Civil War South, see Cornell Dolan (2014).
[19]  Hewitt discusses the letter as a medium that attempts to control the reader's interpretation of it, and it converges well with Bodenheimer's account of Eliot's concern with the obstacles the letter faces confronting the receiver's different mind and circumstances, and with the fallibility of communication through letters.

"[o]ur language is a hard one for poetry"—and that she likes prose—"English prose better than poetry"; HBS1). These two moves constitute a departure from the attitude of a humble, indecisive fan who did not gather confidence to write this letter for a year and has to give reasons for approaching the author at all. (My interpretation of this delay as indecisiveness is due to the letter's generic similarity to fan mail rather than to any solid textual or biographical evidence; in reality, Stowe's personal and professional troubles in the preceding year, such as her son Fred's drinking problem, would amply suffice to keep anyone from initiating new epistolary friendships).

In the second half of the letter, with her carefully phrased critique of Eliot, Stowe establishes herself as an equal, a fellow-writer. Of course, she is certain from the outset that she is, to understate, not unknown to Eliot, so she has reason to believe that the gesture of pulling herself up to the stature of an equal should not be surprising or, hopefully, offensive to her addressee. Nevertheless, as I suggested above, such an approach to the author as an equal may be a gesture that lies at the core of the fan letter as a phenomenon. Even a reader who is not herself a writer already makes such a gesture when she "dares" to approach the author directly—and in writing—to express her opinion (even if it is a very favorable, admiring one). Addressing the author directly is the reader's response to, and enactment of the role of a respected and trusted addressee or interlocutor that a novel often casts the reader in.

From giving a piece of advice, Stowe goes on to articulate her critical appraisal of Eliot's work:

> What strikes me most in your writings is the *morale.* You appear to have a peculiar insight into the workings of the moral faculties—and the religious development through all its phases which is very similar to that of Goethe—so complete is the understanding that you seem to have with each phase that one cannot divine which of all that you have drawn is the one with which you most deeply sympathise—but following you through all the lanes & winding high ways & by ways of religious thought one often asks *where* does this pilgrim find *home*?—What is the *rest* of this explorer? I see your footsteps sometimes in places where one is both glad & sorry to see that another has been—glad because the heart always throbs at sympathetic tokens . . . sorry because; there was no water and no rest –". (HBS1, Cognard-Black 25)

What I see here is a very interesting blending of critical authority assumed by Stowe in her pronouncement on Eliot's writer character and specific worth (saying that she is at her most insightful in the matters of ethics and belief), and a different, more intimately friendly attitude or function of reflecting back to Eliot a coherent, meaningful, aesthetically appealing and thus reassuring image of herself. While at first, it seems that Stowe is somewhat confused by the protean (or, to use Keats's similar characterization of Shakespeare, "chameleon"[20]) ability of Eliot's to identify with very different states or stages of belief, eventually Stowe manages to cast the spiritual figure she encounters in these texts into a Romantically appealing coherence: the author as a dauntless lonely traveler, at once a "pilgrim" (devout worshipper and believer) and an "explorer" (modern, intellectual, adventurous, competent), journeying through places that are at once sacred and untraced. (Eliot's reply shows that she received this gesture of Stowe's very gratefully).

What is even more interesting is a corresponding process that I see taking place here regarding Stowe's own sense of self: it seems that in the process of articulating Eliot's authorial figure as she sees it, Stowe finds that this pilgrim's footsteps in the most remote regions mark Stowe's *own* path: it turns out that Stowe, the reader, has been in these regions *before* the author. Thus, along with the excitement at the rare meeting with a similarly motivated soul, she can extend her own compassion to the traveler who found "no water and no rest". This gesture of greeting a fellow spiritual seeker may bear a shade of religious patronizing (positing Stowe the reader as someone who had traveled these regions *before* on her own and presumably did find solace, rest, and water—a shade that eventually recurs in Stowe's later letters to Eliot). More importantly, however, it illustrates how the reader realizes herself, in the

---

[20]　John Keats's letter to Richard Woodhouse, 27 October 1818.

process of describing and reflecting on her reading experience, to be *another* such pilgrim, recognizing in the author figure, as in a mirror, *her own* image. The joyful surprise (being "glad") at this recognition and the lasting significance it seems to have for Stowe, motivating her to reread and annotate the novels and to dwell for a year on the intention to communicate this experience in a letter, point at the deeply personally significant psychological impact of this literary encounter on Stowe as a reader. Another mechanism of producing unique subjective meaning, generation or refining subjectivity that Stowe's letter seems to reflect is the familiar principle of structural differentiation: the work of assuming a combination of definable group identities, defining oneself *as against* various groups of others: I am a woman, not a man; I am American, not British or German or French[21]. This process often, but not necessarily, assigns a negative value to one group and a positive value to the other. Stowe performs this work of fine-tuning her own identity throughout her correspondence with Eliot, but in her initial letter (published in Cognard-Black and Walls) she starts by defining her addressee:

> You are by nature so *thoroughly English*—Your mind, has in the most airy play of its imagination that English definiteness that refuses to exhale in a mist & turn to a mere cloud—so that I cannot believe that you have come into pantheism in the German way—It requires pipes & tobacco & indefinite coffee to bring that about—besides you are as thoroughly a *woman* as you are English. For sometimes I read your writings supposing you man but come to the contrary con clusion [*sic*] from internal evidence No my sister, there are things about us that no *man* can know & consequently no man can write - & being a woman your religion must be different from man's. (HBS1, Cognard-Black 25-6)

Such comments might appear odd and even inappropriate in a letter to a stranger, but it would sound very natural in a close conversation between new friends, who talk about who they are and what it is in them that brings them together. Having thus placed Eliot in a custom-made system of gender and national categories, in which certain mental qualities are conditioned by group identity (English, woman, writer, spiritual seeker), Stowe creates a clear, detailed and fixed image of an other against which she can define herself—mostly by identification, but also by differentiation. In this and later letters, Stowe defines herself as American, belonging to a newer, less cultivated yet more vital and joyful world, one supposedly exotic for her correspondent who belongs to the "Old World." In fact, the process of the reader defining the author and herself by structural differentiation may be ultimately viewed as another kind of mirroring process: Stowe's identification with Eliot regarding most components of her identity leads to an insight about herself as a female writer and religious believer: there are things about us and the way we relate to God, that "no man can write".

George Eliot's reply strikes a variety of related cords and sets the terms on which their subsequent epistolary relationship will be conducted. First, she indicates that Stowe's warm response to her work was not only very welcome but also, to a great extent, healing:

> I value very highly the warrant to call you friend which your letter has given me. . . . [I]t made me almost wish that you could have a momentary vision of the discouragement, nay, paralyzing despondency in which many days of my writing life have been past [sic], in order that you might fully understand the good I find in such sympathy as yours—in such an assurance as you give me that my work has been worth doing. But I will not dwell on any mental sickness of mine. (GE2, Haight V:29)

According to Bodenheimer, expressing insecurity about the worth of her writing is an extremely characteristic "narrative gesture" in Eliot's correspondence and journals, complicated later in life by her internal conflict about being successful. Such recurrence raises a general question about the notion of

---

[21] On Stowe's use of Eliot's work to construct the concept of "strong femininity", see Cognard-Black, "You Are as Thoroughly *Woman* as You Are English: Strong Femininity and the Making of George Eliot", in her *Narrative in the Professional Age* (Cognard-Black (2004)).

"production of subjective meaning" in these interactions: should it be restricted to *discovering* something about oneself, or extended also to cover repeated performance, perhaps with variations, of certain communicative gestures that are known to oneself and others as hallmarks of one's individuality? (There is also the option of becoming aware, for the first time, that this specific gesture one has just performed in the letter is in fact a recurrent and characteristic one, in which case one's own writing serves as a mirror.)

Eliot next fills in for Stowe the circumstances in which the letter was received, and how it confirmed Eliot's first and very favorable glimpse of Stowe as a person, which came from Stowe's letter to her fan Eliza Lee Cabot Follen many years earlier[22]. It may appear surprising that there is no awkwardness in this case about having read a letter addressed to another person. Yet Eliot shows no scruple about mentioning it—the terms in which she does so suggest a general understanding between the three people involved that the letter was public enough to be read by the addressee to a third party. Moreover, the letter had actually been published shortly after Stowe wrote it, in 1852—which demonstrates again how permeable the boundaries were between private correspondence and public/publishable writing, in the framework of parlor literature. Thus Eliot provides Stowe with feedback and reassuring information on how her communication and her personality more generally had been received early on within a particular (and, being Eliot's own, sufficiently elite) circle of British readers.

Another passage, closer to the end of Eliot's first letter to Stowe, constitutes a favorable mirroring of the same kind as we saw in Stowe's initial letter. However, while Eliot's concluding comments stress Stowe's talent as a reader, they might also be interpreted as an attempt to program, control, or direct Stowe's response, to use Elizabeth Hewitt's terms from *Correspondence and American Literature*. Eliot casts Stowe as the perfect reader (specifically, for this letter):

> Letters are necessarily narrow and fragmentary, and when one writes on wide subjects are liable to create more misunderstanding than illumination. But I have little anxiety of that kind in writing to you, dear friend and fellow-labourer—for you have had longer experience than I as a writer, and fuller experience as a woman, since you have borne children and known the mother's history from the beginning. I trust your quick and long-taught mind as an interpreter little liable to mistake me. . . . I must believe that the joyous, tender humor of your books clings about your more immediate life, and makes some of that sunshine for yourself which you have given to us. (GE2, Haight V:31)

Perhaps this letter, apart from offering a highly favorable mirroring of Stowe, has elements of prescriptive interpellation that imposes high expectations and constraints. After all, Stowe's image is constructed here as one of an ideal woman writer and reader—someone who has "known the mother's story from the beginning", and someone who is unboundedly generous, a giver without reckoning, an unfailing source of sunshine for close and distant others. Interestingly, Eliot addresses her as a "fellow labourer", which sounds more prosaic and spatially bounded than the image of a fellow traveler emerging from Stowe's initial letter.

In Eliot's later letters, this representation or invocation of Stowe recurs as a never-failing reader, whose abundant experience of life, combined with benevolence and sympathy, comprises a superb apparatus for reading Eliot's intention correctly (thus, Stowe's role as primarily a reader in this relationship is confirmed by Eliot in her turn). Another recurring sentiment or attitude on Eliot's part is that of indebtedness to Stowe for her expressions of attention and friendship, and Eliot's regret for failing to carry out fully her obligations as a correspondent and to adequately reciprocate Stowe's attention. Stowe, for her part, often sounds apologetic for not being able to refrain from writing again, and at one point pleads that Eliot not take her letter as another duty that would weigh on her. Stowe's third (unpublished) letter opens: "I grieve sincerely that your burden of care was increased by

---

[22] Eliza Lee Cabot Follen (1787–1860), a prominent Bostonian abolitionist, educator and writer, and an author of nursery rhymes, was Eliot's acquaintance.

so much, as any supposed necessity of answering my letter or my husband's [Calvin Stowe enclosed a letter of his own in the same envelope with Stowe's second letter] . . . Don't feel obliged to *acknowledge* and believe that we follow you with loving sympathy . . . "

## 7. "Is There No Magnetic Telegraph for Us?": Envisaging Communication Across All Barriers in Harriet Beecher Stowe's Subsequent Letters

Stowe's second letter to Eliot seems exemplary to me as an epistolary appeal for friendship, and in accordance with the gesture that it is performing, establishing close contact without meeting physically, its central theme is disembodied communication. It begins: "My Dear Friend/ I came home from Canada where I have been getting Engh copyright & find your kind note . . . It is as like you as possible—the *impersonal you*, we, in our house, have known so long". "The impersonal you" is perhaps the best formulation I have come across to describe a reader's retrieval of the speaking author from behind the veil of print (and in fact my own language here echoes "the veil" as a spiritualist term that Stowe uses later in this letter). Complementary to this description of the reader's experience, the better known "kindred hand" passage names and epitomizes what the whole letter performs—*the writer's* activity as a reaching out for contact and understanding[23]: "A book is *a hand* stretched forth in the dark passage of life to see if there is another hand to meet it—Now in your works, if you could read my marked edition of them, you would see how often the hand has met the kindred hand".

I want to end this essay with a closer look at several kinds of disembodied, immaterial, or quasi-immaterial forms of communication or connection that Stowe invokes in this letter: religious belief, sympathy, writing *and* reading of literature, postal communication, telegraph, and spiritualism. The latter is a particularly strong recurring theme in Stowe's letters to Eliot, and its recurrence is likely to seem jarring and inexplicable to anyone who reads the correspondence as a whole: Eliot repeatedly expresses her lack of credence, let alone enthusiasm, for the subject. However, I think that for Stowe, spiritualism serves as the common denominator between all the communication forms on which she relies in her sense of closeness to her friend. Spiritualism epitomizes the motif of disembodied presence—whether of God to the believer, of the spirits of the dead to the living, or of her interlocutor and herself to each other. It seems that Stowe viewed the connection between authors, readers, and characters of literary works as an integral part of this continuity.

In her second letter to Eliot, she discusses her most recent book *Oldtown Folks*, making excited revelations of the close connections of many characters to their prototypes in Stowe's and her husband's life. The protagonist, a young New England boy who sees spirits, is based on Calvin Stowe. Ghosts and apparitions play a crucial role, both in the development of the plot and as part of the moral landscape of the book. These features of the novel make for a natural transition in the letter from discussing the book to writing at great length about the spread of modern spiritualism in America. Stowe harshly criticizes the American scientific establishment and its aggressive attitude of denying spiritualist phenomena—hoping, apparently, for a quite different attitude on Eliot's part than the skeptical one she received.

> We have had a war [the Civil War] that has put almost every family into mourning. Like the old land of Egypt, there is scarce a house where there is not one dead—& hence this sudden increase of spiritualism. It is the throbbing of the severed soul to the part of itself that is gone within the veil—and should be dealt with reverently & sympathetically not brutally In this, as every other vox populi, there is a vox dei, *if* we only can find out what it is For me—my faith is that of *Dinah* [Dinah Morris, an unusual female Methodist preacher in *Adam Bede*] your loveliest & most living creation—*you* who live in so many that as in Shakespeares [sic] case one doesn't know which is *you*" (HBS2)

---

23 On "the kindred hand" passage see (Cognard-Black and Walls 2006, pp. 21–22).

　　　I think there is a significant connection between the beginning and the end of this passage. Here the concept of writerly and readerly imagination is fused with sympathy as empathic substitution, to the point of quasi-physical merging (as in Melville's letter to Hawthorne following the latter's sympathetic critique of *Moby Dick*[24]). In spiritualist communication, various spirits temporarily, partially enter the medium's body, and in the case of a writer, as Stowe outlines it, there seems to be a similar though reversed relation: the writer inhabits, "lives in" other bodies, those of characters, although it is the writer who has a body, while the characters are non-corporeal. A similar pattern informs the phrase "my faith is that of Dinah": Stowe as a reader finds that part of her mind, namely, her faith, is shared by a character in Eliot's novel, and thus it becomes unclear whether she "lives" in the novel or the novel's character "lives" in her. This connection allows Stowe to adopt Dinah as a sign or a symbol for naming part of her mind, namely, her faith—thus becoming part of her subjectivity. The brutally carnal metaphor, "the throbbing of the severed limb to the heart" (suggestive of the reality of battlefield), is replaced with the spiritualist terms "throbbing of the severed soul to the part of itself that is gone within the veil"—a substitution in line with the general drive of the passage towards reworking the trauma of war into a new cultural reality—that of a more humanly useful faith and a new national unity of a more spiritual kind (be it only a unity of grief, "like the old land of Egypt"). This, after all, is yet another kind of disembodied presence of Americans to one another.

　　　Eliot's second letter presents highly skeptical though tactful comments regarding the issue of spiritualism. One might thus expect that Stowe would drop the subject—but she doesn't. In a later letter, an account of a conversation Stowe had with the spirit of Charlotte Bronte meets with Eliot's circumscribed but firm refusal to take Stowe's claims on faith. Even though Eliot was interested in the parascientific discourse on psychism and spiritualism, as evident from her 1859 novella "The Lifted Veil", her evidence-based approach to science is in the final account precisely what Stowe denounces in her letters as insensitive and, as it appears, essentially masculine.

　　　Stowe, by contrast, fuses together spiritualist and idealized technical means of communication in the image of an imaginary telepathic connection—which she calls "magnetic telegraph" early in her third letter: "I love you—& I talk to you sometimes when I am quite alone so earnestly that I should think you must know it even across an ocean—is there no magnetic telegraph for us?". Apparently, she uses the term here not literally but as a metaphor for her strong sense of familiarity and closeness with Eliot's mind, a closeness that comes just short of conducting reciprocal mental conversations with her. Magnetic telegraph was at the time a widely used metaphor for a spiritual or telepathic connection that Stowe wishes for, and this metaphor characteristically casts the supernatural concept of spiritualist contact into an image of an exterior and material or quasi-material, technological channel of mental communication. One spiritualist newspaper in the 1850s was even titled *Spiritual Telegraph*, and the concept of the telegraph in the nineteenth century, based on quasi-material electricity, might appear hardly less mysterious to a layperson (as shown, e.g., by the range of usages of the word "magnetism" from technical/scientific to occult) than the concept of spiritualist contact, which involved another quasi-material (and at the time quasi-scientific) substance of "ectoplasm" that served as a means of communicating with the spirits, and could even be photographed. Thus, in Stowe's letter, "magnetic telegraph" becomes the ultimate technical metaphor for spiritual communication—perhaps an early outlining of a *virtual relationship* as disembodied presence of friends to each other, enabled by technical means. What seems to inform Stowe's relationship with Eliot throughout the correspondence is this sense of virtuality: the possibility of communicating across and in spite of a dividing "veil" or distance, through the mediation of progressive technologies and institutions, be it the mass print, Victorian post, electromagnetic telegraph or a spiritualist medium. The correspondence suggests that all these media, instead of eroding subjective meaning and destroying the traditional social base for meaningful social

---

[24]　See (Silverman 2002).

interaction (including art), as suggested by some nineteenth- and twentieth-century intellectuals (e.g., Walter Benjamin), can be put to use as vehicles for significant relationships that generate such meaning.

## 8. Conclusions

This paper has performed a close reading of the opening of the Stowe–Eliot correspondence for the interpersonal dynamics and rhetorical gestures that established what can be called a close and lasting virtual friendship between the two authors. The way in which the two writers made use of all communicative media available to them at the time constitutes an early example of subsequent uses of and expectations from ever-faster evolving media. The fact that the printing press and the steamer, or even the telegraph, did not destroy the unique subjective meaning of the individual book, writer or reader, but on the contrary, contributed to creating such meaning by enabling strong interpersonal connections at long distance, goes against the deeply ingrained cultural apprehensions about modern technocracy, which have historically accompanied technological progress. While previous feminist research on nineteenth-century American and English women writers and readers revealed their crucial role in shaping middle-class sensibilities and establishing middle-class power, this paper makes a structurally similar argument. As a case study, the virtual friendship between Stowe and Eliot illustrates women writers' and readers' innovative use of new, diverse, and powerful communicative media, including literary publication, to create a deeply meaningful social and interpersonal connection even when physical meeting is impossible.

**Funding:** This research was funded by Israel Science Foundation Post-doctoral Fellowship "Reading and Subjectivity: Authors as Readers". Post-doctoral advisor: Barbara Hochman.

**Acknowledgments:** I gratefully acknowledge Barbara Hochman's truly valuable mentorship and generous support in conducting the research that led up to this publication, and for commenting on early versions of the article. For access to the unpublished letters of Harriet Beecher Stowe, I am indebted to The Henry W. and Albert A. Berg Collection of English and American Literature; The New York Public Library; Astor, Lenox, and Tilden Foundations. My particular thanks to Isaac Gewirtz for handling the publication permission.

**Conflicts of Interest:** The author declares no conflict of interest.

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
