# Peer review of "“The Impersonal You”: Mass Print and Other Communication Technologies in the Virtual Friendship of Harriet Beecher Stowe and George Eliot"

_humanities, doi:10.3390/h9020037_

Round 1

Reviewer 1 Report

Let me commend the author for the many close readings of Stowe’s letters to Eliot—with attention to how Stowe’s word choice (including even crossed-out words) speaks to her self-construction as both a “fan” of Eliot’s novels yet also as a fellow woman writer and well-known authorial figure.  I believe these careful and complex interpretations of Stowe’s tone—and the possible motivations behind her rhetorical choices—are the strongest element of the article as it now stands. 

I also found the author’s overarching argument convincing—specifically, that Stowe doesn’t differentiate between Eliot’s public novels and the didactic narrative voice contained within them and Eliot’s private letters directed to Stowe herself, i.e., that these communications are all on a continuum of intimacy for Stowe, revealing Eliot’s “true nature” a fellow writer and human being (especially as a woman with spiritual-religious feeling).  I believe this dynamic does indeed speak to a particular kind of reading that was prevalent in Victorian Anglo-America, in which readers believed that narrators and characters within mass-market novels were “speaking directly” to them—a dynamic that was also obviously gendered, since the transatlantic novel in English appealed primarily to women from the outset, trading on intimate genres such as journals and personal letters that did some of the cultural work of domesticity and femininity.  Indeed, Stowe mixes up Eliot’s fact with her fiction repeatedly, trying to divine the beliefs and idiosyncrasies of Eliot the-flesh-and-blood-person within her characters and narrators as much as in the words she writes within her private letters.

However, I found the framing of the article less convincing.  Novel-reading is distinct from art-gazing or owning—if nothing else, in terms of the solitary and intensely private nature of reading (vs. viewing art in public or in someone’s parlor), and the significant differences between consuming words vs. consuming images—and so starting the article with Benjamin’s theories on the elision of the original with the reproduction as a result of the rise of mass media felt a bit strained.  Far stronger, I think, would be an introduction that articulates the nuances of novel reading among Anglo-Americans, especially women, of the Victorian period.  As such, I recommend that the author consult Jennifer Phegley’s book, Educating the Proper Woman Reader, and also Nancy Armstrong’s Desire and Domestic Fiction.  (It may be helpful as well to return to foundational work on Stowe as a producer of “parlor literature,” such as Jane Tompkins’ invaluable Sensational Designs.

In fact, I believe the article actually begins with the second paragraph and Eliot’s own concern that her final novel would be nothing more than one more book tossed on the “heap of books” that were being churned out and consumed quickly after the Civil War.

In addition, at the outset of this article, there is an omission in the paragraph in which the author provides a quick run-down of scholarly places where either parts of the Stowe-Eliot correspondence have been published or works in which this correspondence has been used as the basis of mounting various arguments about the nature of transatlantic correspondences among writers and readers in the nineteenth century.  Specifically, I urge the author to consult Cognard-Black’s monograph, Narrative in the Professional Age, particularly Chapter One, “‘You are as Thoroughly Woman as You Are English’: Strong Femininity and the Making of George Eliot.”  In this chapter, I believe the author will find Cognard-Black’s argument important for his/her/their own—particularly the claim that Stowe and her contemporary Elizabeth Stuart Phelps used their readings of George Eliot’s novels to construct what they saw as a universal aesthetic of “strong femininity,” i.e., a cultural ideal that located social, political, and artistic power in the symbol of an exceptional everywoman embodied in the figure of George Eliot.  In other words, this chapter argues that Stowe and Phelps thought Eliot was speaking directly to them—as women and as fellow writers—through her novels.  The section entitled “The Making of George Eliot” might be of particular interest.

I also felt that the author’s glancing gesture to the psychoanalysts Lacan and Winnicott, in lines 330–332, were unearned and undeveloped—not to mention anachronistic in a piece that’s primarily about the cultural context of novel readers and writers in the late 1860s and 1870s.  Thus, I would cut this reference entirely.

As I said above, the “meat” of this article—with its close readings of the letters—is both strong and compelling, but when the author arrived at the conclusion, I felt as though we veered into yet another argument altogether, one that was meant to circle back to the introduction with Benjamin but that didn’t tie together the insights gleaned from the letters themselves.  If the writer wants to end with a synthesis that links Victorian reading practices and engagements of empathy via novels to nineteenth-century communications media, then the whole article needs to be more intentional about considering how the Stowe-Eliot letters participate within the popular media of the moment—mass print, the telegraph—while, at the same time, undermining the impersonal distancing that these technologies facilitate.  But it seems to me that, instead, this piece as drafted demonstrates how novel-reading, novel-writing, and personal correspondence work as a kind of counternarrative to the facelessness of burgeoning modernity ushered in by mass communications technologies.  In other words, it’s Stowe’s belief in, and metaphorical use of, spiritualism that seems to articulate a new “way of reading,” one that turns characters, narrators, the voice found in a letter, the voice heard at a seance, as well as the image of an author-writer inside one’s own mind into a continuous spiritual selfhood.

Finally, if you’ll allow me to be a bit pedantic, there are some inaccuracies in the author’s direct quotations from Stowe’s letters to Eliot, in terms of dropped words, missing italics, incorrect punctuation, etc., from the original manuscripts held by the Berg Collection.  In particular:

—In line 130, the quote “you have said so much in books that you need not write me letters” should be “You have said so much in books that you need not to write me letters”;

—In lines 177-178, the quote “"And now I am beginning to hear from you every month in Harper's. It is as good as a letter” should be “And now I am beginning to hear from you every month in Harper [sic] — & it is as good as a letter”;

—In lines 257–260, the author doesn’t cite where the quote comes from, and, strangely, the author inserts a “[sic]” that shouldn’t be there, since the original does not include the extra “to” that this author has added.  Here is the correct quote from the original:  “But I was at that time heavily taxed writing a story that I am just now with fear & trembling giving to the English world[.]  It is so intensely American that I fear it may not out of my country be understood, but I cast it like a waif on the waters[.]”;

—In the block quote encompassing lines 293 to 303, in line 298, the “and” that the author has inserted between “lanes and winding” should be an ampersand—i.e., “lanes & winding” — and in line 302, it should read:  “tokens sorry because; there was no water and no rest—” (i.e., the author has dropped the semicolon between “because” and “there”);

—In the next block quote encompassing lines 341 to 349, in line 344 the word “it” before “requires” should be capitalized (“It requires”), and in line 347, the “and” between “know” and “consequently” should be another ampersand—i.e., “know & consequently”;

—In line 428, the original letter omits the apostrophe s on “husband’s”—it’s just “husbands”—and in lines 429-430, the author has quoted “Don't feel obliged to acknowledge and believe that we follow you with loving sympathy,” but the original manuscript reads, “Dont feel obliged to acknowledge and believe that we follow you with loving sympathy—”;

—In lines 435–437, the author has quoted this letter as reading “My Dear Friend / I came home from Canada where I have been getting English copyright and find your kind note It is as like you as possible – the impersonal you, we, in our house, have known so long,” but the original reads, “My Dear Friend / I came home from Canada where I have been getting Engh copyright & find your kind note—It is as like you as possible—the impersonal you, we, in our house, have known so long” (i.e. “Engh” not “English” and a missing dash and ampersand as well as underlinings where, before, the author used italics; the same issue w/ italics vs. underlinings happens in line 442 as well);

—In lines 473–476, “if” should be italicized (“if we only can”) as should “Dinah” (“that of Dinah”), and the “and” between “loveliest” and “most living” should be an ampersand (i.e., “loveliest & most living”); also, “you” should be italicized in line 475 (“you who live in so many”) and again in line 476 (“one doesn’t know which is you”); and, finally,

—In lines 501–503, what’s currently quoted— “I love you – and I talk to you sometimes when I am quite alone so earnestly that I should think you must know it even across an ocean – is there no magnetic telegraph for us?” should instead read as this:  “I love you—& talk to you sometimes when I am quite alone so earnestly that I should think you must know it even across an ocean—is there no magnetic telegraph for us?”

Author Response

Many thanks to Reviewer 1 for the invaluable comments and suggestions.  Following them, I have reframed the article, placing it explicitly within the tradition of the feminist historians of reading, where it in fact belongs. Thus, I have added a new opening to the intro and a corresponding conclusion section. Particular thanks for the Cognard-Black reference, "The Making of George Eliot," have incorporated it in the review of research on the correspondence. Have cut out the reference to psychonalytic theories. Also many thanks for the corrections in the quotes, have introduced them throughout. All changes can be seen in Track Changes mode.

Reviewer 2 Report

The author is well informed on the subject of letters and fan mail, and particularly attentive to the dynamics of the fan letter that claims relationship, even parity or superiority, with the writer addressed. The essay astutely examines the interesting correspondence, and brings to bear the existing scholarship on it, placing it in the context of forms of communication at that time and now. The author is also conversant with theoretical concepts, though these are introduced in a heavy manner. Generally, the article is a worthy reconsideration of the intimacy created in the two authors’ correspondence, though it is not as clear as it should be as to how it adds to existing scholarship on these texts, in spite of a very clear summary of previous scholarship. I recommend that the author revise the text, primarily at the beginning and the conclusion, to highlight its main contributions, reorganize often loose sentence structures, and remove the occasional casual diction. Without saying that “I” is out of place in such an essay, in this version of this very good scholarly and critical piece the “I will show” or “I think” moments detract a bit. The subheadings don’t quite do all the work of pacing and organizing the essay, and there needs to be a more substantial final paragraph(s) to highlight the original observation about the two authors’ use of the technology of letters to adjust their personae as readers/peer writers.

As to the obtrusive use of theory: the opening gambit of Walter Benjamin seems unnecessary; reduce this and place it below an introduction that focuses on epistolarity and the production of inter-subjectivity—the hope that two persons can meld through a medium—more than on authenticity/reproduction. Similarly, around lines 330-32, the sentence citing Lacan and Winnicott seems cumbersome as phrased; it’s quite true that mirroring is relevant to the recognition of Eliot’s spiritual quest in Stowe’s first letter, but most readers can get this conceptual reference more swiftly. Especially delete “mechanisms...here.”  The idea of structural differentiation, which should be contrasted with mirroring even though it is similarly a sighting of the Other, is interesting, but again, lines 333-36 can be much condensed.

The author can consider the matter of technology and science further. It is brought in at first, but not sustained. Stowe toured the UK, which is partly how Eliot’s mutual friend met her, I think. But GE never traveled to the US. Was this possibly because she could not give a lecture tour, as a woman living with a married man?  Eliot was very involved in Lewes’s scientific research, and friends with advanced theorists of social laws and evolution, but they were both mainly in the age of rail and steam, not telegraph. Clearly, the postal service was vastly improved for that brief period when a letter could be sent in London in the morning and reach another part of London a few hours later.  This is relevant to the essay, but it needs to be more clearly followed through.

The theme of production of subjectivity is fresh in the close reading of the correspondence, circa lines 375-382. It surfaces almost as if this essay’s author is writing to a friend, and could be reframed somewhat less volubly for greater effect.

When mentioning spiritualism and the veil, it is crucial that the essay note that George Eliot published “The Lifted Veil,” which includes a sense of magnetism and new science as well. Stowe would have found many UK writers more inclined to spiritualism than Eliot, including E Barrett Browning, the Brontes, etc. “Magnetism” is an idea that predates 1860s, related to Galvanism, and occurs in Mary Shelley to an extent, but certainly in C. Bronte. Can the author highlight the tension between Stowe’s US version of spiritualism and GE’s European rationalism?

Is this author aware that some critics have attended to the influence of Stowe’s Dred on Daniel Deronda? There is more uncanny effect in DD than in most of GE’s previous fiction.

When noting that Stowe writes from her orange grove, do note that the Stowes did not themselves plant this (as owners, they ordered it to be planted); it is painful to contemplate a plantation in Florida right after the Civil War. Labor conditions in citrus groves in Florida continued to be vassal-like, forced labor well into 20th c. Without being expert on this, I would urge finding a source that studies the irony of Stowe in Reconstruction and post-Reconstruction South.

Some smaller points:

closer=more closely

examine all the uses of “mass”—as print runs do vary, and some consider mass markets to be a later 19th c. development (certainly for periodicals)

around line 61 and 81, the same fact about Cognard-Black's publication is repeated.

Reduce the detailing in prose of the history of discussion of the correspondence of GE and Stowe.

What is “framework of parlor literature”?

“lack of credit, leave alone”: lack of credit isn’t clear in this context—her lack of credence, perhaps? It should be “let alone”

Author Response

Many thanks to Reviewer 2 for the thorough reading and invaluable comments and suggestions.

I have reframed the article with a new intro opening and conclusion. The subsection titles were extended and rewritten, some new section divisions added, for better structure and readability. The use of first person has been reduced, style changed to a more formal register. 

References have been added to all the issues raised by the reviewer (Eliot's "The Lifted Veil" and the two authors' different approaches to spiritualism, the influence of Dred on Daniel Deronda, the complicated political aspects of owning an orange plantation in Florida after the Civil War), which definitely must improve the quality of the paper. 

The suggested technical corrections have been introduced.

All changes can be seen in Track Changes mode.

Reviewer 3 Report

Thank you for asking me to review this engaging essay. I enjoyed reading it and I think it makes an interesting and original contribution to scholarship on correspondence networks and women's writing.

In my opinion it could be more effectively organised, if it began with the correspondence that is the focus of the essay, rather than Walter Benjamin. I think it is likely readers will come to this essay for the letters, and not the theoretical approach that shapes the argument. That being said I think the Benjamin works well, so what I am talking about is a matter of prioritisation. I would recommend describing the correspondence, and the particular issues it raises, then introducing Benjamin as a theoretical means of unlocking that particular problem, and revealing a fresh perspective on the correspondence.

Throughout the writer uses "I" a little too much for my taste. In particular I would recommend opening paragraphs by more clearly signposting how the paragraph advances the argument, as beginning with an "I" sentence qualifies the impact. Also I should note that the piece moves between "I" and "one" uneasily, and to a native speaker this sounds odd.

I would recommend removing the reference on p. 10 to Lacanian psychoanalysis as this is a complex idea, and should either be more fully developed or dropped. As it seems like a side issue to the overall discussion as it stands, I would recommend a cut.

Throughout the English does not read colloquially. There is some awkward phrasing ('this gesture of Stowe's' for example instead of simply 'Stowe's gesture') and missing words. 

Author Response

I am grateful to Reviewer 3 for the valuable comments and suggestions.

I have followed the suggestion on restructuring the introduction, adding a more relevant research context and then introducing Benjamin as a familiar and thematically relevant "they say" to my own claim. Also the division into subsections has been elaborated for clearer structure, and a formal conlcusion has been added.

The references to psychoanalytic theory have been removed, and the language revised and formalized throughout.

All changes can be seen in "Track Changes" mode.